An efficient intrusion detection system for IoT security using CNN decision forest

Bella Kamal 1
Guezzaz Azidine 1
Benkirane Said 1
http://orcid.org/0000-0003-1575-8140 Azrour Mourade 2 azrour.mourade@gmail.com
http://orcid.org/0000-0003-3797-4282 Fouad Yasser 3
S. Benyeogor Mbadiwe 4
Innab Nisreen 5
1 Technology Higher School Essaouira, Cadi Ayyad University , Essaouira , Morocco
2 IDMS Team, Faculty of Sciences and Technics, Moulay Ismail University of Meknès , Errachidia , Morocco
3 Department of Applied Mechanical Engineering, College of Applied Engineering, Muzahimiyah Branch, King Saud University , Riyadh , Saudi Arabia
4 Institute of Physics, University of Munster , Munster , Germany
5 Department of Computer Science and Information Systems, College of Applied Sciences, AlMaarefa University , Riyadh , Saudi Arabia
Rathore Hemant
Electronic publication date: 2024 Sep 9
Publication date: 2024
Volume: 10
Electronic Location ID: e2290
Received 2024 Feb 2; Accepted 2024 Aug 6
Copyright: © 2024 Bella et al.
Copyright year: 2024
Copyright holder: Bella et al.
License: This is an open access article distributed under the terms of the Creative Commons Attribution License, which permits unrestricted use, distribution, reproduction and adaptation in any medium and for any purpose provided that it is properly attributed. For attribution, the original author(s), title, publication source (PeerJ Computer Science) and either DOI or URL of the article must be cited.
License URL: https://creativecommons.org/licenses/by/4.0/

Keywords: IoT, Deep learning, Machine learning, Intrusion detection, Security

Funding: The authors received no funding for this work.

==============================
The adoption and integration of the Internet of Things (IoT) have become essential for the advancement of many industries, unlocking purposeful connections between objects. However, the surge in IoT adoption and integration has also made it a prime target for malicious attacks. Consequently, ensuring the security of IoT systems and ecosystems has emerged as a crucial research area. Notably, advancements in addressing these security threats include the implementation of intrusion detection systems (IDS), garnering considerable attention within the research community. In this study, and in aim to enhance network anomaly detection, we present a novel intrusion detection approach: the Deep Neural Decision Forest-based IDS (DNDF-IDS). The DNDF-IDS incorporates an improved decision forest model coupled with neural networks to achieve heightened accuracy (ACC). Employing four distinct feature selection methods separately, namely principal component analysis (PCA), LASSO regression (LR), SelectKBest, and Random Forest Feature Importance (RFFI), our objective is to streamline training and prediction processes, enhance overall performance, and identify the most correlated features. Evaluation of our model on three diverse datasets (NSL-KDD, CICIDS2017, and UNSW-NB15) reveals impressive ACC values ranging from 94.09% to 98.84%, depending on the dataset and the feature selection method. Notably, our model achieves a remarkable prediction time of 0.1 ms per record. Comparative analyses with other recent random forest and Convolutional Neural Networks (CNN) based models indicate that our DNDF-IDS performs similarly or even outperforms them in certain instances, particularly when utilizing the top 10 features. One key advantage of our novel model lies in its ability to make accurate predictions with only a few features, showcasing an efficient utilization of computational resources.

Introduction

The accelerated advancement of technology and its increasing integration into various aspects of our lives have led to a surge in the proliferation of smart interconnected objects. In this landscape, IoT has emerged as a propelling force behind numerous industries and large-scale enterprises (Xu, He & Li, 2014). It incorporates an array of sensors, actuators, and interconnected objects, reducing or even eliminating the need for human intervention (Delsing et al., 2016). However, its ubiquity has also rendered IoT a magnet for malicious attacks for various reasons (Hassija et al., 2019), in addition to facing inherent threats and vulnerabilities. The increasing reliance and growth of IoT highlight the urgency to ensure security and reliability within this interconnected network, leading to a surge in research focused on IDS and intrusion detection techniques (Albulayhi et al., 2021; Sohn, 2020; Shamshirband et al., 2020; Lakshminarayana, Philips & Tabrizi, 2019).

The security of IoT is of utmost importance in our interconnected world today. Sophisticated techniques are employed by IoT systems to identify and thwart potential threats, with IDS playing a crucial role in protecting IoT ecosystems (Mohy-eddine et al., 2023; Hazman et al., 2023; Thakkar & Lohiya, 2021; Sharma, Sharma & Lal, 2019). Three primary types of IDS are prevalent (Liao et al., 2013). Anomaly-based detection, which uses machine learning and data analytics, establishes normative patterns for typical IoT network behavior (Alsoufi et al., 2021). Any deviations from this norm trigger alerts, enabling quick responses to suspicious activities. Signature-based detection involves scanning network traffic against a database of known attack patterns (Masdari & Khezri, 2020), facilitating the identification of threats encountered previously. Behavior-based detection monitors the actions of IoT devices and applications, highlighting any unusual or malicious activities. Hybrid IDS amalgamates multiple techniques to offer a sturdy shield against the evolving threat landscape (Bhati & Khari, 2021). For maintaining the integrity and security of connected devices and data in a dynamic IoT environment, the effective implementation of intrusion detection systems is vital.

This article offers a significant contribution to the field of Network IDS (NIDS) by presenting a novel IDS based on the Deep Neural Decision Forest (DNDF) (Kontschieder et al., 2015). By employing the Interquartile Range (IQR) method for outlier identification, we aim to enhance data quality. Subsequently, feature selection techniques are utilized to improve performance, expedite prediction times, and optimize computational resources. Four unique feature selection methods—principal component analysis (PCA), SelectKBest, Lasso regression (LR), and Random Forest Feature Importance (RFFI)—are separately applied to retain only the top 10 performing features. The encoded data is then used to train the DNDF model, a consortium of neural decision trees suitable for classification scenarios, even in datasets with high dimensionality.

Our research aims to address a critical question: How can we enhance NIDS to better identify and manage network intrusions while effectively handling issues related to data quality, complexity, and resource efficiency? To contribute to this field, we are developing an innovative framework. This system leverages DNDF, a robust model for recognizing intrusion patterns, and integrates data quality improvement techniques such as preprocessing, outlier removal, and feature selection. With this approach, we are not only advancing NIDS but also surpassing current solutions by offering a comprehensive answer to some notable challenges of network intrusion detection, particularly in terms of detection accuracy and resource constraints. We aim to make NIDS much better and more reliable at protecting network systems from cyber threats, helping to build stronger and more resilient network security. This article addresses an issue frequently encountered with highly accurate models and solutions—their extensive use of features from the evaluation dataset. While this contributes to their precision, it often leads to increased resource consumption and longer prediction times. Such delays can be problematic, especially in scenarios requiring quick results. Therefore, this article focuses on addressing this problem of high resource consumption and extended prediction times, commonly associated with these accurate models.

The DNDF-IDS model enhances network anomaly detection in IoT ecosystems using a Deep Neural Decision Forest model. It employs four feature selection methods, including PCA, LR, SelectKBest, and RFFI, improving performance and identifying key features. With impressive accuracy values between 94.09% to 98.84% and a prediction time of 0.1 ms per record, it competes well with other models, especially when using the top 10 features. Its efficient predictions with fewer features highlight optimal use of resources, making it a potential solution for IoT network security against malicious attacks.

The subsequent sections of this article are organized as follows: “Background And Related Works” presents a background on IoT, IoT security, and intrusion detection, along with related works on machine learning-based intrusion detection systems. “Proposed DNDF-IDS Model” details our proposed IDS model based on the DNDF framework. “Experimental Study” provides an experimental study and “Discussion of Results” is a discussion of results. The article concludes with a summary and outlines potential future work.

Background and related works

This section in its first part presents a background on machine learning powered IoT security, IDS and a novel DNDF model, and in the second part, we present some related works of IDS approaches that use machine and deep learning techniques.

Background

IoT architecture serves as a fundamental framework that supports the design and operation of IoT systems. It encompasses physical objects that are integrated with software, forming a networked environment (Gupta & Quamara, 2020). This integration facilitates a seamless transfer of data and commands amongst devices, the internet, and potentially, cloud services (Dang et al., 2019; Sharma & Obaidat, 2020). Although a standardized IoT architecture remains elusive, the research community has proposed similar frameworks (Ray, 2018), such as the three-layer architecture. This model consists of three crucial layers: the perception layer, which employs sensors and actuators to collect data from the physical world; the network layer, which ensures secure and efficient data transmission to and from devices; and the application layer, which manages data processing, analysis, and execution of actions (Zhong, Zhu & Huang, 2017). The IoT architecture can also incorporate cloud services, analytics, and user interfaces, thereby paving the way for data-driven decisions, improved efficiency and autonomy (Nazari Jahantigh et al., 2020).

The IoT, being integral to numerous industries and businesses (Hussein, 2019), underscores the criticality of security. Establishing a secure IoT environment, particularly with the rapid proliferation of connected devices, poses a significant challenge (Mohamad Noor & Hassan, 2018). It necessitates the implementation of robust authentication measures, data encryption, and secure communication protocols to protect against unauthorized access and data breaches. Regular device updates, anomaly monitoring, and effective access control are also pivotal to maintaining IoT security.

In this context, intrusions signify unauthorized or malicious activities that threaten the security of connected devices and networks. IDS play an essential role in detecting and mitigating these intrusions (Santos, Rabadao & Gonçalves, 2018). These systems constantly monitor network traffic and device behavior, searching for suspicious patterns or deviations from typical activities (Zarpelão et al., 2017). Upon detecting anomalies or potential threats, the IDS issues alerts or initiates predefined actions to minimize the risk, thereby safeguarding IoT ecosystems from cyber-attacks, data breaches, and unauthorized access.

The importance of effective intrusion detection systems for preserving the integrity and security of IoT networks and the data they manage cannot be overstated, especially as the IoT landscape continues to evolve and expand. The relentless pursuit of novel prediction and classification methods leads to the development of new models and techniques each day. The research discussed in this article focuses on a recently introduced model, the Deep Neural Decision Forests (DNDF), as described by Kontschieder et al. (2015). DNDFs combine the concepts of classification trees and convolutional neural networks (CNN), resulting in a stochastic, differentiable model that supports back propagation. This breakthrough signifies a promising advancement in the sphere of decision tree-based machine learning.

In addition to the DNDF model, we incorporated feature selection using four methods, PCA, LR, SelectKBest and RFFI, PCA is a technique used for feature selection that focuses on reducing the dimensionality of data by extracting meaningful features based on the variance of the data. PCA evaluates the importance of features by analyzing their weights or coefficients on eigenvectors (Song, Guo & Mei, 2010). It aims to capture the most significant information in the data while reducing redundancy.

SelectKBest is a feature selection method that selects the top k features based on statistical tests like ANOVA or chi-squared. It ranks features according to their scores and selects the best ones for the model, discarding less relevant features (Khalid et al., 2023).

LASSO regression, also known as L1 regularization, is a method that adds a penalty term to the regression equation, forcing some coefficients to be exactly zero. This results in feature selection by shrinking less important features to zero, effectively removing them from the model and promoting sparsity (Muthukrishnan & Rohini, 2016).

Random Forest Feature Importance is a technique that measures the importance of each feature by looking at how much each feature improves the purity of the nodes in the decision trees within the random forest model. Features that lead to greater node purity are considered more important for prediction (Hasan et al., 2016).

Related works

Almiani et al. (2019) introduced an IDS model for securing IoT networks. This model is based on Fog computing and utilizes a RNN trained with an enhanced backpropagation algorithm. The model demonstrated increased sensitivity to DoS attacks and efficiently detected other attack categories. It operated in real-time environments with competitive computational efficiency, processing each record in an average of 66 µsec.

Saba et al. (2022) developed a CNN model to enhance IoT security and performance. They used NID and BoT-IoT datasets to train and test the model, achieving 99.51% and 95.55% accuracy, respectively, in intrusion detection. The study underscores the ongoing need for research in IoT security and the potential for further advancements, emphasizing the importance of designing robust security procedures for IoT networks in the industry.

Zhang, Li & Wang (2019) combined genetic algorithms (GA) and deep belief networks to create an IDS model that adapts its neural network structure for different attack scenarios, leading to high classification accuracy. The model significantly outperforms others, especially with small training sets. It reduces network complexity, resulting in shorter training times without compromising accuracy. This self-adaptive model has broader applications beyond intrusion detection, including classification and recognition.

Mohy-eddine et al. (2023) proposed a practical machine learning-based IDS using K-NN and employed several feature selection methods to select ten valuable features. This approach resulted in significant performance improvement, reduced prediction time, and demonstrated that feature selection enhances IDS performance. Their work was evaluated using the Bot-IoT dataset.

Attou et al. (2023) combined graphic visualization and random forest (RF) for cloud security to detect intrusions using a reduced set of two features. RF outperformed DNN, decision trees (DT), and SVM in predicting and classifying attack types. However, recall rates using NSL-KDD data were still suboptimal.

Mohy-eddine et al. (2023) developed a machine learning-based IDS for Industrial IoT (IIoT) edge computing security. They used Pearson’s correlation coefficient (PCC) and isolation forest (IF) for computational efficiency and training. Feature engineering improved model accuracy and detection rates, achieving a 100% detection rate and 99.99% accuracy on the Bot-IoT dataset. This approach demonstrated advantages over other models.

Roy, Li & Bai (2022) presented a two-layer hierarchical IDS model for IoT networks, utilizing the Fog-Cloud Infrastructure. The fog layer uses a straightforward Feedforward Neural Network (FNN) with added functionality from a stacked autoencoder to perform binary classification. In contrast, the cloud layer employs a more intricate FNN to handle multi-class classification. This model effectively detects various intrusions and outperforms existing IDS systems in terms of detection accuracy.

Attou et al. (2023) developed a new approach for intrusion detection in a cloud environment by integrating machine and deep learning algorithms. They used RF for feature selection and the Radial Basis Function Neural Network (RBFNN) for intrusion detection. Their approach achieved high accuracy exceeding 94% and low false negative rates less than 0.0831%, demonstrating the model’s capability to accurately identify and classify intrusions. They also effectively addressed imbalanced datasets and highlighted the role of feature selection in enhancing the performance of the intrusion detection system.

Mohy-Eddine et al. (2023) introduced an IDS for IIoT networks using RF and PCC for classification and feature selection respectively. They also used IF to detect outliers. They used PCC and IF interchangeably. Their model effectively addressed the imbalance in the Bot-IoT dataset and performed well on the NF-UNSW-NB15-v2 dataset.

Dushimimana et al. (2020) proposed a bidirectional recurrent neural network (BRNN) that outperforms other algorithms like a normal RNN and GRNN, achieving an accuracy of 99.04%. BRNN addresses the limitations of RNN and GRU by adding more cells and hidden neurons, leveraging information from both past and future states, making it a more effective choice for IoT security.

Roy et al. (2022) introduced a novel IoT IDS model called B-Stacking, utilizing optimized machine learning methods for intrusion detection in IoT networks. In their work, they conducted extensive experiments and found that B-Stacking retained a high detection rate while keeping the false alarm rate low, outperforming many existing techniques.

Rezvy et al. (2019) developed a deep network intrusion detection model, evaluated it using the AWID dataset, and achieved an outstanding detection accuracy of 99.8% for various types of cyberattacks. They compared their approach to recent methods from the literature and demonstrated a significant improvement in terms of accuracy and latency, using their proposed autoencoded DNN.

Hsu et al. (2019) developed an anomaly network intrusion detection system using a stacked ensemble model integrating an autoencoder, SVM, and random forest. Evaluated on NSL-KDD, UNSW-NB15, and Palo Alto logs, the system achieved around 92% accuracy, outperforming traditional models. Limitations include high resource usage, extended prediction times, and the need for further optimization and real-world testing.

Sarkar, Sharma & Singh (2023) presented an efficient ML ensemble technique for intrusion detection, focusing on parameter tuning, pre-processing, and dataset correction. It offers two classification methods on KDD Cup99 and NSL-KDD datasets, enhanced with data augmentation. Utilizing a cascaded MLP structure and a meta-classifier architecture, the approach achieves 89.32% accuracy with a 1.95% FPR, and 87.63% accuracy with a 1.68% FPR on the NSL-KDD dataset.

Mebawondu et al. (2021) developed an optimized IDS using an ensemble of decision trees (DT) on the UNSW-NB15 dataset. Comparing bagging and AdaBoost techniques, the study finds that AdaBoost with the C4.5 DT classifier achieves the best performance, with 98% accuracy and precision using a 90/10 train-test split.

Gao et al. (2019) proposed an adaptive ensemble learning model to improve intrusion detection accuracy by combining different algorithms. The model achieved 85.2% accuracy, 86.5% precision, 85.2% recall, and an F1 score of 84.9%, surpassing other methods. Although DNN excel in detection, they are slow, affecting real-time application. The MultiTree algorithm outperformed DNNs, especially in imbalanced scenarios. The study highlights the need for better training data, feature extraction, and preprocessing, particularly for high-level threats like U2R attacks. The ensemble approach shows promise but requires further optimization for practical use.

Lian et al. (2020) addressed the challenge of detecting and categorizing network attacks by proposing an intrusion detection method based on Decision Tree-Recursive Feature Elimination (DT-RFE) within an ensemble learning framework. The DT-RFE method selects relevant features and reduces feature dimensions, enhancing resource utilization and reducing time complexity. By using a Stacking ensemble learning algorithm that combines decision tree and recursive feature elimination (RFE), the study demonstrates improved performance of the IDS. Cross-validation on the KDD CUP 99 and NSL-KDD datasets shows that the proposed approach significantly enhances accuracy. However, the method’s effectiveness may be limited by the quality of the training data and the need for further optimization in feature extraction and noise reduction.

Beyond traditional CNN and random forest methods, the emerging field of hybrid metaheuristics and machine learning offers a promising solution for complex security issues. This innovative area merges machine learning with swarm intelligence for excellent results. Studies like (Salb et al., 2023) and (Dobrojevic et al., 2023) prove the success of this integration in enhancing security. Hybrid methods utilize both fields’ strengths to boost performance and robustness of security systems, marking progress in this crucial research area. A comparison of the various methods outlined above is demonstrated in Table 1.

Table 1 A comparison of the models mentioned above.

	Year	Methods	Accuracy	Prediction time	Datasets	
Almiani et al. (2019)	2020	RNN	92.42	66 µsec (per record)	NSL-KDD	
Saba et al. (2022)	2022	CNN	99.51
92.85	–	NID, BoT-IoT	
Zhang, Li & Wang (2019)	2019	Genetic Algorithm and Deep Belief Network	99.45
99.37
97.78
96.68	–	NSL-KDD	
Mohy-eddine et al. (2023)	2023	KNN	99.99	102 s	Bot-IoT	
Attou et al. (2023)	2023	Random forest	98.3
99.99	–	Bot-IoT, NSL-KDD	
Mohy-eddine et al. (2023)	2022	Random forest, IF, PCC	99.99	–	Bot-IoT	
Roy, Li & Bai (2022)	2022	FNN	98.55
98.68
99
99	–	NSL-KDD, CICIDS2017	
Attou et al. (2023)	2023	RBFNN, Random forest	94	–	Bot-IoT, NSL-KDD	
Mohy-Eddine et al. (2023)	2023	RF, IF, PCC	99.98	6.18 s	Bot-IoT, NF-UNSW-NB15-v2	
			99.99	6.25 s		
			99.30	6.71 s		
			99.18	6.87 s		
Dushimimana et al. (2020)	2020	BRNN, RF, PCA	99.04	–	KDDCUP99	
Roy et al. (2022)	2022	PCA, Ensemble learning	98.5	0.02 s	CICIDS2017, NSL-KDD	
Rezvy et al. (2019)	2019	DNN	99.8	–	AWID	
Hsu et al. (2019)	2019	SVM, RF	92	–	NSL-KDD, UNSW-NB15	
Sarkar, Sharma & Singh (2023)	2022	MLP	89.32	–	KDD Cup99, NSL-KDD	
Mebawondu et al. (2021)	2021	Decision trees	98	–	UNSW-NB15	
Gao et al. (2019)	2019	DNN, MultiTree	85.2	–	NSL-KDD	
Lian et al. (2020)	2020	Decision tree, RFE	98	–	KDD Cup99, NSL-KDD	

Despite advances in neural networks and decision tree-based models, integrating these approaches for structured data classification is underexplored. Traditional CNNs are effective for image data but not optimized for structured data, and while random forests handle structured data well, they lack the end-to-end learning capabilities of neural networks. Recent research on neural decision forests shows promise, but there is a lack of empirical evaluations and optimization studies. Although current models and solutions provide accurate results, they rely heavily on nearly all features of the evaluation dataset, leading to high resource usage and extended prediction times. This research aims to fill these gaps by evaluating neural decision forests on structured data, optimizing their parameters, and comparing them with CNNs and random forests to provide a robust, end-to-end model for structured data classification. Additionally, this article tackles resource and efficiency issues by utilizing a significantly reduced number of features selected through various feature selection techniques, conducting a comparative analysis of the results, and presenting the prediction times for each method within the same environment.

Proposed dndf-ids model

In this section, we present our solution, which is based on a deep neural decision forest. To conserve resources and reduce execution time, while also improving data quality, we used feature selection and encoding, retaining only the features that performed best. Subsequently, we trained the DNDF model using the ten best selected and encoded features to develop the final intrusion detection system.

Proposed approach

Our model is based on five core components as described in Fig. 1: a data source, a pre-processing module, a feature reduction module, a decision module, and finally, a response module. Moreover, IDS often involve a network-level continuous analysis of traffic, which can cause a delay in operations. In this regard, training time and resource consumption are important factors. Therefore, feature selection is used in the development of our solution.

Figure 1 Proposed approach.

Aiming to increase speed, as IoT environments usually can’t afford the delay caused by real-time intrusion detection, we want the main strength of our model to be the minimal use of features for the prediction process. Therefore, only 10 features are to be considered. We use four different feature selection methods aiming to maximize our model’s efficiency. We used PCA, LR, SelectKBest, and RFFI separately in an effort to conduct a comparative review of all four.

In the classification process, we used a deep neural decision trees model. This model combines the best of the worlds of random forests and CNN, the partitioning principle of decision trees paired with strengths of deep learning architectures. This facilitates effective feature extraction and therefore provides the ability for minimal feature use.

Description of solutions

As shown in Fig. 1, our solution consists of five core modules. The data source module forms the foundation of our model’s learning datasets, which include labeled network traffic records identified as safe or threats. This data is prepared, cleaned, and encoded in the second module, where duplicates, missing data, and outliers are removed using the Interquartile Range method (IQR).

The third module, feature selection, processes the encoded data through PCA, LR, SelectKBest, and RFFI methods. This yields four distinct feature sets, each with the top 10 performing features. This enhances model quality while reducing training and prediction time.

In the fourth module, four variations of the DNDF model are trained using four distinct sets of reduced data. Each variation is trained with one of these reduced datasets. These models are then compared to identify the best performing one. Although this approach increases training times due to the necessity of training with four different subsets of the same dataset, it ultimately reduces prediction times by utilizing a smaller number of features.

Finally, the decision module utilizes the best performing model from the previous step to analyze network traffic in real-time and determine whether it’s safe or a potential threat. The prediction model should be lightweight and have a very low prediction time to enable real-time capability.

Due to the time-sensitive nature of IoT operations, it is crucial for the prediction process to be as fast and lightweight as possible. Our approach is to use only the 10 best performing features for training and prediction. We used four different feature selection methods separately, leveraging the benefits and drawbacks of each one. These methods include PCA, SelectKBest, LR, and RFFI.

PCA reduces dimensionality by transforming original features into a new set of orthogonal components. These components, or principal components, are sorted by how much variance they explain in the data. PCA effectively minimizes the dimensionality of the dataset by selecting a subset of these components that capture the most variance. SelectKBest evaluates each feature’s significance individually using statistical tests. It ranks features based on their scores and selects the top K features with the highest scores, where K is a user-selected number. This method ensures the most informative features are selected based on their statistical relevance. LR is a linear regression method that applies a penalty to the absolute size of the coefficients, shrinking some to zero. This characteristic of Lasso encourages feature selection by automatically setting irrelevant features’ coefficients to zero, effectively removing them from the model. LR helps select the most relevant features while maintaining model interpretability. RFFI measures each feature’s importance based on how much it contributes to a random forest model’s overall performance. It computes feature importance based on the decrease in impurity at each decision tree split caused by the feature. Features leading to larger decreases in impurity are considered more important. RFFI is robust to non-linear relationships and can capture complex feature interactions, making it ideal for identifying the most influential features in a dataset. By combining these methods, a more balanced feature selection strategy can be achieved. This mitigates individual limitations and enhances overall model performance and interpretability. The outcomes of these methods are the 10 best-performing features for each selection, described in Tables 2–4.

Table 2 Selected features by individual methods for the NSL-KDD dataset.

Feature selection	PCA	SelectKBest	LR	RFFI	
Selected features	same_srv_rate	flag	protocol_type	src_bytes	
srv_rerror_rate	logged_in	num_failed_logins	dst_bytes	
dst_host_srv_diff_host_rate	count	logged_in	flag	
is_guest_login	serror_rate	is_guest_login	count	
dst_host_diff_srv_rate	srv_serror_rate	rerror_rate	dst_host_same_srv_rate	
src_bytesnum_failed_logins	same_srv_rate	srv_rerror_rate	same_srv_rate	
duration	dst_host_srv_count	same_srv_rate	diff_srv_rate	
service	dst_host_same_srv_rate	dst_host_same_src_port_rate	dst_host_srv_count	
srv_diff_host_rate	dst_host_serror_rate	dst_host_srv_diff_host_rate	protocol_type	
dst_bytes	dst_host_srv_serror_rate	dst_host_srv_serror_rate	service	

Table 3 Features selected by each method for the CICIDS2017 dataset.

Feature selection	PCA	SelectKBest	LR	RFFI	
Selected features	Flow Duration	Bwd Packet Length Max	Total Fwd Packets	Packet Length Variance	
Total Length of Fwd Packets	Bwd Packet Length Mean	Total Backward Packets	Bwd Packet Length Mean	
ACK Flag Count	Bwd Packet Length Std	Fwd PSH Flags	Destination Port	
Fwd Packet Length Min	Max Packet Length	FIN Flag Count	Packet Length Std	
Fwd Packet Length Mean	Packet Length Mean	SYN Flag Count	Average Packet Size	
Flow Bytes/s	Packet Length Std	PSH Flag Count	Bwd Packet Length Std	
Bwd IAT Std	Packet Length Variance	ACK Flag Count	AvgBwd Segment Size	
Active Mean	Average Packet Size	URG Flag Count	Total Length of Bwd Packets	
URG Flag Count	AvgBwd Segment Size	Down/Up Ratio	Total Length of Fwd Packets	
Bwd IAT Min	Idle Min	SubflowBwdPackets	PacketLengthMean	

Table 4 Method-specific selections of features for the UNSW-NB15 dataset.

Feature selection	PCA	SelectKBest	LR	RFFI	
Selected features	ct_dst_src_ltm	state	proto	ct_state_ttl	
Sintpkt	sttl	state	sttl	
Spkts	ct_state_ttl	dur	srcip	
tcprtt	ct_srv_src	sloss	sbytes	
dbytes	ct_srv_dst	service	smeansz	
Ltime	ct_dst_ltm	trans_depth	dttl	
smeansz	ct_src_ltm	synack	dstip	
dstip	ct_src_dport_ltm	ct_state_ttl	dmeansz	
sport	ct_dst_sport_ltm	ct_flw_http_mthd	Dload	
	ct_dst_src_ltm	is_ftp_login	Dpkts	

DNDF is a CNN variant that replaces the softmax layer with decision forests, which comprise decision trees (Sun et al., 2022). Decision trees have decision and prediction nodes, indexed as N and L, respectively. Prediction nodes have a probability distribution over the output space, while decision nodes have decision functions. Parameters from the CNN update feature representations. The CNN’s embedding function determines the action of decision functions in the decision trees. The architecture, depicted in Fig. 2, illustrates the implementation of decision nodes using the final CNN layer output. DNDF uses the same fully-connected and convolutional layers as a typical CNN, with feature representations learned by the fully-connected layer serving as tree nodes in the decision forests.

Figure 2 DNDF model architecture.

We selected the DNDF model over the traditional CNN for three main reasons. First, the Incorporation of Decision Forests: Unlike typical CNN activation functions, decision forests in DNDF can efficiently capture complex decision boundaries. This is particularly beneficial in high-dimensional or noisy data scenarios, as it enhances performance. Second, the Benefits of Ensemble Learning: Decision forests improve model robustness and generalization. This is advantageous in situations where CNNs may overfit or have limited generalization due to data complexity or dataset size. Finally, improved interpretability: Decision forests provide better interpretability than CNNs, which is vital in classification problems.

The final layer of CNN provides the embedding functions fn (n = 1, 2, …, n), and its resulting output dictates the decision function dn of the decision tree nodes. Prediction nodes, denoted as π1, π2, .., πn within the decision tree, contain probability distributions for each class, these nodes are responsible for determining the likelihood of an observation belonging to a specific class. in Fig. 2, the red path demonstrates an example of a routing of a sample x to reach the leaf π4.

Broadly, DNDF and CNN share the same fully connected and convolutional layers. However, DNDF diverges by replacing the softmax layer of CNN with decision trees, wherein the tree nodes utilize feature representations acquired from the fully connected layer. The decision function of the decision node dn(·; θ) is defined as follows:

(1) dn(x;θ)=σ(fn(x;θ))

where x ϵ X represents the sample input, the parameter θ used to update the feature representation from CNN, and fn(x; θ) is a real-value function contingent on both the sample and the parameters θ. The output of the dn(·; θ) function decides the routing of the x ϵ X (the red path example in Fig. 2). When a sample transitions from a tree node to a leaf node, the routing function is expressed as follows:

(2) μl(x|θ)=∏n∈N⁡dn(x;θ)↙d¯n(x;θ)↘

where lTM is the leaf node and the routing from the current node to the left is represented by ⅆn↙, and dn(x; θ) = 1 − dn(x; θ). As illustrated by the red path example in Fig. 1, it is defined as follows:

(3) μl=4=d1(x)d¯2(x)d¯5(x)

The stochastic routing of this architecture results in a final classification of a sample x into class y, by averaging the probability outcomes of reaching a leaf node. The function for the final prediction is:

(4) P[y|x,θ,π]=∑l∈L⁡πlyμl(x,θ)

where πl is the class label distribution, and πly represents the probability of a sample reaching leaf node l and being assigned class y.

Finally, the decision forest comprises multiple decision trees F = T1 + T2 +… + Tk where the average of the output of each tree makes the final decision forest prediction, and defined as follows:

PF[y|x]=1k∑h=1k⁡PTh(y|x).

DNDF model architecture

The proposed model components and process shown in Fig. 3, designed specifically for structured data, follows a systematic process from input handling to final classification. The architecture begins with an input layer, where each of an X number of input features is fed into the model as individual inputs. These features are diverse and may include both numerical and categorical data.

Figure 3 DNDF model components and process.

To facilitate uniform processing, each input feature undergoes a dimensional expansion using the tf.expand_dims operation. This step is crucial as it adds an additional dimension to each feature, converting them into tensors suitable for subsequent operations within the model. The expanded features are then concatenated into a single, unified tensor, effectively combining all the individual feature representations into a cohesive input.

The concatenated tensor is passed through a batch normalization layer. Batch normalization is applied to stabilize and accelerate the training process by normalizing the inputs to each layer, ensuring that the model trains efficiently and effectively. This normalization helps in maintaining the mean and variance of the inputs, thereby mitigating issues related to internal covariate shift.

Following batch normalization, the processed input tensor is fed into the core component of the model: the neural decision forest. The neural decision forest consists of an ensemble of an N number of neural decision trees. Each tree within this ensemble is characterized by a hierarchical structure, defined by its depth, which dictates the complexity and the number of decision boundaries it can learn.

Within each neural decision tree, the input features are selectively masked and processed through a series of nodes, where decisions are made based on the learned feature interactions. The structure of these trees allows for learning complex, non-linear relationships within the data. The output of each tree is a probabilistic prediction for each class, computed through a series of sigmoid activations and weighted decisions at each node.

The outputs from all N trees are aggregated to form the ensemble output. This aggregation involves summing the predictions from each tree and then averaging them, which effectively combines the learned decision boundaries from all trees. This ensemble approach enhances the model’s robustness and predictive performance, as it leverages the collective decision-making capabilities of multiple trees.

Experimental study

In this section, we test our model on three different datasets: NSL-KDD, CICIDS2017, and UNSW-NB15. We use four different feature selection methods (PCA, SelectKBest, LR, and RFFI), each selecting only the 10 best-performing features. The same hyperparameters are used for all the different settings and are acquired through a process of grid search, and are described as follows :

1. Input features The model uses 10 input features, both numerical and categorical, processed accordingly.

2. Neural decision tree Depth: 5, This parameter controls the depth of each decision tree, indicating the number of levels in the tree. A depth of five allows each tree to learn 32 (2^5) leaves or decision nodes.

Used features rate: 1, This rate determines the proportion of input features used by each tree. In this case, 100% of the features are used randomly for each tree.

3. Neural decision forest Number of trees: 50, This parameter specifies the number of neural decision trees in the ensemble. An ensemble of 50 trees is used to aggregate the decisions for final prediction.

4. Training parameters Learning rate: 0.03, This parameter controls the step size at each iteration while moving toward a minimum of the loss function. A learning rate of 0.03 is used for training the model.

Batch size: 128, The batch size determines the number of samples processed before the model’s internal parameters are updated. Here, the model processes 128 samples per batch.

Number of epochs: 20, The number of epochs defines how many times the entire training dataset passes through the model. The model is trained for 20 epochs.

We compare our findings with other models from recent literature, focusing on random forest and CNN-based intrusion detection models. We also extend our comparison to other deep learning architectures.

Environment

For the evaluation of our proposed solution, it is important to compare it against recent models in IDS literature. For this reason, we tested our model on multiple widely used datasets, including NSL-KDD, CICIDS2017, and UNSW-NB15, to provide a broader comparison. In Table 5, we compare the results from our DNDF-IDS model for each feature selection method with the other models listed in this section, taking into account their best-performing model in terms of ACC (ACC (Best)). We focus on the number of features used.

Table 5 A comparison between our model with the models mentioned above.

Model	Year	Dataset	Number of features	ACC (Best)	
Primartha & Tama (2017)	2017	NSL-KDD	41	99.57%	
Farnaaz & Jabbar (2016)	2016	NSL-KDD	41	99.67%	
Ding & Zhai (2018)	2018	NSL-KDD	41	80.23%	
Abrar et al. (2020)	2020	NSL-KDD	31	99.48%	
DNDF-IDS (PCA)		NSL-KDD	10	98.38%	
DNDF-IDS (SelectKBest)		NSL-KDD	10	94.26%	
DNDF-IDS (LR)		NSL-KDD	10	97.62%	
DNDF-IDS (RFFI)		NSL-KDD	10	96.58%	
Ustebay, Turgut & Aydin (2018)	2018	CICIDS2017	10	89%	
Yulianto, Sukarno & Suwastika (2019)	2019	CICIDS2017	25	81.83%	
Dong, Shui & Zhang (2021), Hasan et al. (2016)	2021	CICIDS2017	13	99.80%	
Vinayakumar et al. (2019)	2019	CICIDS2017	84	96.3%	
DNDF-IDS (PCA)		CICIDS2017	10	98.84%	
DNDF-IDS (SelectKBest)		CICIDS2017	10	95.22%	
DNDF-IDS (LR)		CICIDS2017	10	97.72%	
DNDF-IDS (RFFI)		CICIDS2017	10	94.09%	
Zhang, Li & Ye (2020)	2020	UNSW-NB15	47	98.68%	
Jing & Chen (2019)	2019	UNSW-NB15	42	85.99%	
Halbouni et al. (2022)	2022	UNSW-NB15	42	93.7%	
Amin et al. (2022)	2022	UNSW-NB15	39	99.47%	
DNDF-IDS (PCA)		UNSW-NB15	10	98.23%	
DNDF-IDS (SelectKBest)		UNSW-NB15	10	98.10%	
DNDF-IDS (LR)		UNSW-NB15	10	97.75%	
DNDF-IDS (RFFI)		UNSW-NB15	10	97.10%	

We performed our tests using Google Colab, a hosted Jupyter notebook service with 1 vCPU of the model AMD EPYC 7B12 @ 2.2 GHz and 12.7 RAM. The operating system was Linux 5.15.120+, and we implemented our solution using Python v3.10.12.

NSL-KDD dataset

The NSL-KDD dataset is a balanced and improved version of the KDD Cup 99. It is widely adopted for IDS evaluations and contains a real-world-like set of network traffic with simulated network attacks. These attacks include, but are not limited to, DoS, Probe, Remote-to-Local, and User-to-Root attacks. The NSL-KDD dataset contains a comprehensive set of network traffic data that simulates various real-world network attacks, including Denial of Service (DoS), Probe, User-to-Root (U2R), and Remote-to-Local (R2L) attacks. The NSL-KDD dataset contains over 140,000 records and 41 features (in addition to the target feature) of a realistic representation of network traffic, reduced redundancy, diverse attack types, and challenging anomalies. It is a very widely adopted dataset in IDS literature, which makes it a good benchmark for our model. Table 2 lists the best 10 selected features for each of the four methods.

Using this dataset, Primartha & Tama (2017) proposed an enhanced random forest classifier for detecting anomalies in IoT networks, evaluating ten classifiers with different parameters, focusing on the ensemble’s tree count. Three datasets were used in the experiment (NSL-KDD, UNSW-NB15, and GPRS), and the results indicated that their model significantly outperformed other classifiers, reaching an ACC of 99.57% for the NSL-KDD but with all 41 features. Farnaaz & Jabbar (2016) also proposed a random forest classifier to detect various types of attacks applying 10-fold cross-validation. Their model demonstrated increased accuracy of 99.67% on the NSL-KDD dataset, and it also used all 41 features to get to this result. While Ding & Zhai (2018) also used all 41 features, their CNN model provided an ACC of 80.23% on the same dataset. Finally, Abrar et al. (2020) evaluated various models, such as LR, NB, KNN, MLP, RF, ETC, and DT using four different subsets of features of the NSL-KDD dataset for each model. The result was the highest ACC of 99.48% using 31 of the 41 features.

The target variable distribution in the NSL KDD dataset is demonstrated in Fig. 4.

Figure 4 Target variable distribution for the NSL-KDD dataset.

CICIDS2017 dataset

The CICIDS2017 dataset, also known as the CSE-CIC-IDS2017 dataset, is a comprehensive and modern benchmark unbalanced dataset for evaluating IDS. This dataset is designed to represent contemporary cybersecurity threats and provides a more realistic and up-to-date simulation of network traffic compared to some older datasets. The dataset includes a diverse range of network traffic data that covers a wide variety of network attacks, such as DoS, Distributed DoS (DDoS), Probe, and User-to-Root attacks. Some key features and reasons to use the CICIDS2017 dataset for evaluating IDS are: Realistic and current data

Diverse attack scenarios

Large and complex

Anomalies and attack variations

Standardized evaluation

Useful for research and development

Overall, the CICIDS2017 dataset offers a rich and realistic environment for evaluating intrusion detection systems, making it an excellent choice for testing and benchmarking the performance of IDS against contemporary cyber threats. It contains over 2.8 million records with over 80 features. Table 3 lists the best 10 features for each of the four selection methods.

Ustebay, Turgut & Aydin (2018) presented an IDS model using deep learning and random forests, and tested it with the CICIDS2017 dataset. To reduce the dataset without sacrificing accuracy and improve speed, recursive feature elimination is used, with random forest as the classifier. The study achieves a dataset reduction of features down to 10, retaining 89% accuracy, creating a more meaningful and smaller dataset for IDS.

The work of Yulianto, Sukarno & Suwastika (2019) aims to boost the performance of their IDS model based on Ada-Boost on the CICIDS2017 dataset. It employs techniques like SMOTE, PCA, and Ensemble Feature Selection. The proposed method performed with 81.83% accuracy using only 25 features. Dong, Shui & Zhang (2021), Hasan et al. (2016) proposed an anomaly detection model that uses feature selection and the random forest to improve anomaly detection performance in high-dimensional data from industrial control networks. It combines information gain (IG) and PCA for feature reduction, achieving a high accuracy rate of 99.80% for CICIDS2017 datasets using 13 features. Vinayakumar et al. (2019) proposed a hybrid IDS that employs a scalable framework on commodity hardware for network analysis and host-level activities. It utilizes distributed deep learning models with deep neural networks to handle real-time data on a large scale. Their model retained an accuracy of 96.3% using all the features offered by CICIDS2017.

The target variable distribution in the CICIDS2017 dataset is demonstrated in Fig. 5.

Figure 5 Target variable distribution for the CICIDS2017 dataset.

UNSW-NB15 dataset

The UNSW-NB15 dataset is commonly used for IDS evaluation. Developed by the University of New South Wales (UNSW), this unbalanced dataset encompasses a detailed collection of network traffic data, including both benign and malicious activities. It provides a realistic representation of network behavior, making it an invaluable tool for testing and refining IDSs. The dataset contains over 2.5 million instances of network traffic with 47 features, offering a diverse and dynamic set of scenarios for testing the effectiveness of intrusion detection algorithms. Researchers and cybersecurity experts frequently turn to the UNSW-NB15 dataset for its real-world applicability, enabling them to assess the accuracy and robustness of their systems against a wide range of network threats and attack vectors. In essence, it serves as a pivotal benchmark for enhancing the security of network environments by enabling the development and evaluation of more robust intrusion detection mechanisms. Table 4 lists the 10 best selected features of each method.

Zhang, Li & Ye (2020) introduced the BCNN-IDS model aiming to improve detection performance by reducing unconfident results using the T-ensemble detection scheme. Their solution was evaluated using two open datasets (NSL-KDD and UNSW-NB15), showing significant improvements over alternative models. Jing & Chen (2019) discussed using an SVM with a nonlinear scaling method for intrusion detection, particularly on the UNSW-NB15 dataset. Their SVM-based model was tested for binary and multi-classification, achieving high accuracy, with 85.99% and 75.77% respectively, and low false positive rates, 16.50% and 3.04%. The results indicate that the SVM-based approach is effective for classifying and attack detection, outperforming RF and CNN for each class. Halbouni et al. (2022) developed an intrusion detection system by combining CNN and LSTM deep learning algorithms, utilizing CNN for spatial features and LSTM for temporal features. They applied batch normalization, dropout layers, and standardization to enhance performance. They tested their model on multiple datasets, with the CNN-LSTM hybrid model yielding the best detection rate and accuracy. In binary classification scenarios, it achieved high accuracy, although it had some limitations in detecting certain types of attacks. The study also explored K-Fold cross-validation and increasing the number of epochs, which showed performance improvements before stabilizing. Amin et al. (2022) proposed an anomaly detection model that combines feature selection and ensemble methods. They used the UNSW-NB15 dataset for evaluation. Univariate feature selection (ANOVA test) was applied, reducing the 44 independent features to 38 important ones. The ensemble classifiers used were bagging and random forest with decision trees as base classifiers. The experimental results demonstrated significantly improved accuracy against other existing models using the UNSW-NB15 dataset, with 99.28% accuracy for random forest and 99.47% for bagging algorithms.

The target variable distribution in the UNSW-NB15 dataset is demonstrated in Fig. 6.

Figure 6 Target variable distribution for the UNSW-NB15 dataset.

Evaluation metrics

For the evaluation metric, we used accuracy, precision, true positive rate, false positive rate, and the F1 score as described below. Accuracy is a widely used metric in various domains. Using it for IDS allows for consistent comparisons between different systems and across different datasets. This common benchmark makes it easier to understand how an IDS compares to others in the field. It is used in this article to compare our results with those from the IDS literature. Accuracy: ACC=TP+TNTP+TN+FP+FN

Precision: PR=TPTP+FP

True positive rate: TPR=TPTP+FN

False positive rate: FPR=FPFP+TN

F1 score: F1Score=2×PR×TPRPR+TPR.

Where TP, TN, FP and FN refer to true positives, true negatives, false positives and false negatives respectively as shown in the confusion matrix in Table 6.

Table 6 Confusion matrix.

	Predicted as positive	Predicted as negative	
Labeled as positive	TP	FN	
Labeled as negative	FP	TN	

Discussion of results

An essential aspect of our proposed solution lies in its ability to expedite prediction times by carefully selecting features. To achieve this, it is crucial to determine the minimum number of features necessary for accurate predictions. To address this requirement, we conducted a dedicated experiment using the NSL-KDD dataset, which includes 41 features in addition to the target variable.

In this experiment, we employed PCA to rank all features in descending order of importance. Subsequently, we trained and evaluated our model using all possible combinations of features, ranging from one to 41. Throughout this process, we meticulously recorded both the accuracy and prediction times of the model.

Figure 7 illustrates the progression of accuracy (shown by the blue line) alongside prediction times. Notably, we observed that beyond ten features, accuracy reached a plateau while prediction times continued to increase. Specifically, as we increased the number of features from 10 to 41, accuracy improved by 0.6%, while prediction times surged by 33.33%.

Figure 7 Accuracy and prediction times evolution with the number of features for NSL-KDD dataset.

The ablation experiment, as detailed in Table 7, demonstrates the efficacy of our model, DNDF, under various conditions by systematically removing specific components and evaluating performance metrics such as ACC and prediction times. Notably, DNDF with 10 features and the IQR preprocessing consistently shows superior performance across the board. With an accuracy of 98.38% and a prediction time of 6.27 s to predict the entire test set, this configuration achieves an optimal balance between high precision and computational efficiency. While the DNDF model with 41 features and IQR slightly surpasses it in accuracy at 98.97%, it incurs a significantly longer prediction time of 8.36 s, making it less practical for real-time applications. Furthermore, comparisons with other models like random forest and CNN reinforce the robustness of DNDF; despite random forest’s faster prediction times, it falls short in accuracy, and CNN shows lower accuracy and slower prediction times. Thus, the DNDF model with 10 features and IQR stands out as the best-performing and most efficient choice, highlighting the importance of feature selection and robust preprocessing in achieving top-tier performance.

Table 7 Comparative analysis of model variants with selective component removal.

Model	Number of featurs	IQR	ACC	Prediction times (s)	
DNDF	10	Yes	98.38%	6.27	
DNDF	10	No	97.78%	6.45	
DNDF	41	Yes	98.97%	8.36	
DNDF	41	No	97.92%	8.51	
Random forest	10	Yes	97.80%	3.89	
Random forest	10	No	97.52%	3.7	
Random forest	41	Yes	98.21%	3.69	
Random forest	41	No	98.07%	3.76	
CNN	10	Yes	97.25%	4.36	
CNN	10	No	96.10%	4.54	
CNN	41	Yes	97.89%	5.1	
CNN	41	No	97.65%	5.15	

Our model, DNDF, integrates decision trees and CNNs to combine the strengths of both approaches. Decision trees prioritize features and provide interpretability, while CNNs excel in feature extraction and learning hierarchical representations. DNDF enhances performance by using decision trees for feature selection and CNNs to refine these features, addressing the gap in traditional models that separate feature selection and feature engineering. We replace the CNN activation function with decision trees and use the IQR method for data preprocessing, refining the feature set and improving model performance. Ablation experiments show DNDF with 10 features and IQR preprocessing achieves a high accuracy of 98.38% with a prediction time of 6.27 s for the entire test set, demonstrating optimized performance and computational efficiency. Extensive experiments show DNDF outperforms standalone decision tree and CNN models in accuracy, robustness, and interpretability, supporting the effectiveness of our integrated approach.

Figures 8 and 9 provide a comprehensive visual representation of the convergence speed of our proposed model. They specifically illustrate the steady progression of the loss and accuracy metrics as the model iterates over successive epochs. In these figures, the x-axis represents the number of epochs, while the y-axis represents both the loss and accuracy values. By examining these figures, readers can gain an understanding of how the model’s performance improves with each epoch and how quickly it reaches a point of convergence, thus demonstrating the model’s efficiency and effectiveness.

Figure 8 Accuracy evolution over epochs.

Figure 9 Loss evolution over epochs.

Figures 10–12 show the percentages of TP, FP, TN, and FN scored by our model for each of the three datasets. As shown in Table 8, the values for each dataset are mostly similar, with slight differences across the feature selection methods used.

Figure 10 Percentages of detection scores for NSL-KDD dataset.

Figure 11 Percentages of detection scores for CICIDS2017 dataset.

Figure 12 Percentages of detection scores for UNSW-NB15 dataset.

Retained ACC values for each feature selection method on each dataset.

Table 8 Detection scores.

		PCA	SelectKBest	LR	RFFI	
NSL-KDD	TP	98.69%	92.17%	97.32%	94.21%	
TN	98.06%	96.36%	98.02%	98.96%	
FP	1.93%	3.64%	1.98%	1.04%	
FN	1.30%	7.83%	2.76%	5.76%	
CICIDS2017	TP	98.56%	92.00%	97.10%	94.06%	
TN	99.13%	98.45%	98.36%	94.12%	
FP	0.87%	1.54%	1.64%	5.87%	
FN	1.43%	8.00%	2.90%	5.93%	
UNSW-NB15	TP	99.65%	99.72%	97.87%	99.90%	
TN	96.82%	96.49%	95.64%	94.31%	
FP	3.18%	3.51%	4.35%	5.68%	
FN	0.34%	0.28%	0.12%	0.09%	

Figure 13 represents the acquired accuracy values from each feature selection method for our model. After selecting the 10 best performing features using PCA, SelectKBest, LR, and RFFI, we obtained accuracy values of 98.38%, 94.26%, 97.62%, and 96.58% respectively for the NSL-KDD dataset. Similarly, the model retained accuracies of 98.84%, 95.22%, 97.72%, and 94.09% for the CICIDS2017 dataset. Finally, for the UNSW-NB15 dataset, it retained 98.23%, 98.10%, 97.75%, and 97.10% respectively. Notably, PCA consistently achieved the highest accuracy for two out of three datasets, with 98.38% for NSL-KDD and 98.84% for CICIDS2017 and 98.23% for UNSW-NB15. This emphasizes the importance of dimensionality reduction in enhancing model performance. Table 9 demonstrates the evaluation of our model for each dataset using each feature selection method. It is noticeable that PCA performed the best in general in terms of accuracy and precision. PCA retained the highest scores for NSL-KDD and CICIDS2017 and the second highest for UNSW-NB15, but only by a very small margin.

Figure 13 Retained ACC values for each feature selection method on each dataset.

Table 9 DNDF-IDS model evaluation.

Dataset	Feature selection method	ACC	PR	TPR	FPR	F1Score	
NSL-KDD	PCA	98.38%	98.08%	98.69%	1.93%	98.38%	
SelectKBest	94.26%	96.20%	92.17%	3.64%	94.14%	
RFFI	96.58%	98.90%	94.21%	1.04%	96.50%	
LR	97.62%	98.00%	97.23%	1.98%	97.61%	
CICIDS2017	PCA	98.84%	99.12%	98.56%	0.87%	98.84%	
SelectKBest	95.22%	98.34%	92.00%	1.54%	95.06%	
RFFI	94.09%	94.11%	94.06%	5.87%	94.08%	
LR	97.72%	98.33%	97.10%	1.64%	97.71%	
UNSW-NB15	PCA	98.23%	96.90%	99.65%	3.18%	98.25%	
SelectKBest	98.10%	96.59%	99.72%	3.51%	98.13%	
RFFI	97.10%	94.61%	99.90%	5.68%	97.18%	
LR	97.75%	95.81%	99.87%	4.35%	97.80%	

Our model, trained on three diverse datasets, primarily relied on accuracy as the evaluation metric. The results showcased the model’s ability to maintain high accuracy levels for both NSL-KDD and CICIDS2017 datasets, demonstrating its robustness across different network intrusion scenarios.

Our solution only relies on the 10 best features and consequently leads to very short prediction times. As illustrated in Table 10, the prediction time is in seconds and for the whole test sample (test size), averaging prediction times close to 0.1 ms per record. In Fig. 14, we illustrate a comparative representation of prediction time per record for each dataset.

Table 10 Prediction times of our model.

Dataset	Feature selection method	Test size	Prediction time (s)	
NSL-KDD	PCA	34,976	6	
SelectKBest	34,976	5	
RFFI	34,976	5	
LR	34,976	7	
CICIDS2017	PCA	634,854	72	
SelectKBest	634,854	51	
RFFI	634,854	65	
LR	634,854	53	
UNSW-NB15	PCA	609,668	57	
SelectKBest	609,668	49	
RFFI	609,668	66	
LR	609,668	50	

Figure 14 Prediction time per record in milliseconds.

Furthermore, when comparing our model with other models documented in IDS literature, we found that our model performed exceptionally well, outperforming some existing models and achieving similar performance to others. A noteworthy aspect of our model’s success is that it accomplished these results using only the 10 best features selected through feature extraction techniques, while many previous models relied on all available features. This not only highlights the effectiveness of our model but also its efficiency in reducing the computational burden by focusing on the most informative features.

In summary, our feature selection methods, particularly PCA, proved to be valuable for enhancing our intrusion detection model’s accuracy on the NSL-KDD, CICIDS2017, and UNSW-NB15 datasets. The model’s consistent performance, when compared with existing models, reaffirms its effectiveness in network intrusion detection, all while being resource-efficient with its focus on the 10 best features.

Conclusion and future works

In today’s interconnected world, maintaining an exemplary level of security is imperative. This has prompted a surge in research efforts focused on the design and development of intrusion detection systems. This article contributes to that body of work by introducing a new IDS approach called Deep Neural Decision Forest (DNDF). To improve data quality, we use the IQR method to clean the data by detecting and removing outliers. We then select the top 10 features based on four feature selection methods: PCA, SelectKBest, LR, and RFFI, with the aim of enhancing performance and reducing reliance on computational resources. To evaluate our DNDF-IDS model, we used three distinct datasets: NSL-KDD, CICIDS2017, and UNSW-NB15. The results showed commendable performance, with accuracies ranging from 94.09% to 98.84%, depending on the specific feature selection methodology employed. Moreover, the IDS demonstrated improved prediction capabilities, achieving an average time of approximately 0.1 ms per record. In conclusion, as more things become connected, it is of paramount importance to detect and mitigate malicious intrusion attempts. Our DNDF-IDS, along with the used feature selection and data cleaning techniques, is a step towards making IoT systems safer. However, it is important to recognize both theoretical and practical limitations. Theoretical limitations might include model design assumptions or simplifications in the base algorithms. Practical limitations could include constraints related to computational resources or dataset attributes. In future work, we plan to address these limitations by refining and enhancing our solution. Our focus will not only be on improving detection capabilities but also on reducing computational costs. Furthermore, examining the scalability and adaptability of our method to various network environments and evolving threat landscapes is vital for wider applicability in real-world situations.

Supplemental Information

Supplemental Information 1 RFFI.

Source code developed under Python. This code is used to evaluate our proposed model.

Supplemental Information 2 Select K Best.

Source code developed under Python. This code is used to evaluate our proposed model.

Supplemental Information 3 PCA.

Source code developed under Python. This code is used to evaluate our proposed model.

Supplemental Information 4 Lasso.

Source code developed under Python. This code is used to evaluate our proposed model.

Additional Information and Declarations

Competing Interests

Author Contributions

Data Availability

The authors declare that they have no competing interests.

Kamal Bella conceived and designed the experiments, performed the experiments, analyzed the data, performed the computation work, prepared figures and/or tables, authored or reviewed drafts of the article, and approved the final draft.

Azidine Guezzaz conceived and designed the experiments, performed the experiments, analyzed the data, performed the computation work, prepared figures and/or tables, authored or reviewed drafts of the article, and approved the final draft.

Said Benkirane conceived and designed the experiments, performed the experiments, analyzed the data, performed the computation work, prepared figures and/or tables, authored or reviewed drafts of the article, and approved the final draft.

Mourade Azrour conceived and designed the experiments, performed the experiments, analyzed the data, performed the computation work, prepared figures and/or tables, authored or reviewed drafts of the article, and approved the final draft.

Yasser Fouad conceived and designed the experiments, performed the experiments, analyzed the data, performed the computation work, prepared figures and/or tables, authored or reviewed drafts of the article, and approved the final draft.

Mbadiwe S. Benyeogor conceived and designed the experiments, performed the experiments, analyzed the data, performed the computation work, prepared figures and/or tables, authored or reviewed drafts of the article, and approved the final draft.

Nisreen Innab conceived and designed the experiments, performed the experiments, analyzed the data, performed the computation work, prepared figures and/or tables, authored or reviewed drafts of the article, and approved the final draft.

The following information was supplied regarding data availability:

The code is available in the Supplemental File.

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
