# Peer review of "An efficient intrusion detection system for IoT security using CNN decision forest"

_PeerJ Computer Science, doi:10.7717/peerj-cs.2290_

## Round 0.1 · original submission · Major Revisions

All three reviewers have raised important points and suggested major corrections. The authors need to incorporate them for paper acceptance.

Reviewer 2 has suggested that you cite specific references. You are welcome to add it/them if you believe they are relevant. However, you are not required to include these citations, and if you do not include them, this will not influence my decision.

**Language Note:** PeerJ staff have identified that the English language needs to be improved. When you prepare your next revision, please either (i) have a colleague who is proficient in English and familiar with the subject matter review your manuscript, or (ii) contact a professional editing service to review your manuscript. PeerJ can provide language editing services - you can contact us at [email protected] for pricing (be sure to provide your manuscript number and title). – PeerJ Staff

Reviewer 1 ·

Basic reporting

Why was DNDF suggested as a variant of CNN instead of CNN in the study? What would be the results if CNN were applied instead of DNDF? The superiority of DNDF over CNN should be presented in comparison with the performance results.

How the hyperparameters of DNDF are obtained should be presented in detail. Why were 10 epochs used? Why was batch size 128 chosen? Why was learning rate 0.01 chosen? Why was num_trees chosen as 25? ...etc. A detailed explanation should be made about hyperparameter selections.

What would the performance results be like if all the attributes in the datasets were used instead of 10 features? It should be given comparatively. The advantages of choosing 10 features should be explained according to the performance results.

What would be the result if different machine learning methods were used instead of DNDF? should be presented comparatively.

It should be explained in detail how feature selection is made with PCA, which is used as a feature reduction/transformation method.

The number of samples in the datasets and their distribution according to classes should be detailed. Performances should be evaluated separately according to the classes (anomaly/benign) in the datasets.

For datasets with unbalanced data distribution, it is necessary to explain in detail whether there is bias or overfitting in the performance results and what precautions are taken for these situations. It should be explained how training and testing data were determined.

Experimental design

How the hyperparameters of DNDF are obtained should be presented in detail. Why were 10 epochs used? Why was batch size 128 chosen? Why was learning rate 0.01 chosen? Why was num_trees chosen as 25? ...etc. A detailed explanation should be made about hyperparameter selections.

Validity of the findings

What would be the result if different machine learning methods were used instead of DNDF? should be presented comparatively.

Reviewer 2 ·

Basic reporting

Dear Authors,
although proposed manuscript has merits, there are some issues that need to be addressed.

My observations are below.

TITLE
Avoid rather unknown acronyms or unspecific wordings like “novel/new”, “improved/enhanced”, “hybrid”, ”fast”, "advanced". Try to avoid overlength, “spaghetti” structure, so, make it short and concise, [ specific comments: Please simplify paper title, focus on Computing Methodology and Studied problem.]

Abstract
Method names should not be capitalized. Moreover, it is not the best practice to employ abbreviations in the abstract, they should be used when the term is introduced for the first time.

Introduction
Introduction should be clearly presented to highlight main ideas and motivation behind the proposed research. Please include and clearly state research question and contributions of proposed study in Introduction. Also, please clearly explain what is "beyond state-of-the-art" in the proposed study.

Literature review
Literature review should be improved to include prospective research field - hybrid methods between metaheuristics and machine learning. Consider using the following papers to enhance the literature review (you may just say that the novel research field successfully combines machine learning and swarm intelligence approaches and proved to be able to obtain outstanding results in different areas of security):

https://www.mdpi.com/2076-3417/13/23/12687
https://www.sciencedirect.com/science/article/pii/S1877050923000145
https://peerj.com/articles/cs-1405/

Experimental design

Please provide detailed flow-chart diagram of proposed methodology.
Also, please provide more details of devised method, so that other authors can easily replicate produced outcomes and methodology.

Visualization of results should be improved - consider using box and whiskers diagrams, swarm plots, etc. Also, please include convergence speed graphs, since the convergence is very important indicator of ML/DL performance especially during the training phase.

More details regarding utilized dataset should be provided, e.g. visual overview of distribution per classes, whether the dataset is balanced, imbalanced, partially balanced, etc. In case of imbalanced datasets, kohen cappa coefficient should be included as metrics for comparative analysis. Precision, recall and f1-score indicators should be provided per each class.

Validity of the findings

More SOTA baseline methods should be included in comparative analysis. Also, ML/DL tuned methods with metaheuristics or with grid search should be encompassed in analysis.

To prove the significance of obtained results, statistical tests must be conducted. There are many statistical tests appropriate for validating results, please choose some tests from the following reference: https://www.sciencedirect.com/science/article/pii/S2210650211000034

Additional comments

Conclusion should be extended to include more details regarding the future work and limitations of proposed study. Limitations should be distinguished between theoretical and practical.

Some references are missing parts, such as pp., publisher, year, etc.

There are some English language and technical errors, please revise, e.g. tables must be aligned on the center of the page, all symbols in equations must be defined, etc.

Reviewer 3 ·

Basic reporting

1. whilst the manuscript structure conforms to Peerj guidelines and introduces the reader with research context, the ‘introduction’ section lacks concrete details such as,
a. Briefly highlight the gap in the existing literature and the need for the proposed model for intrusion detection
b. How the proposed model is designed to address the gap.
c. how the proposed system is validated with respect to the ablation experiments and datasets
d. Importantly, the key contribution / novelty of the proposed system in comparison to existing state-of-the art methods should be summarized to seek the attention of the readers
2. The authors are advised to briefly discuss about the four feature selection methods and deep learning technique that are leveraged in designing the proposed intrusion detection system under background section.
3. Related Works Section:
a. lacks logical flow and makes difficult for the reader to understand the existing research gap in the field of the study. For example, there are literature that discusses about intrusion detection in Fog, IOT and cloud rather than literature related to feature selection pertaining to intrusion detection system.
b. The authors are advised to narrow down the relevant literature and then summarize the gap in coherent manner to enhance the reader understanding on the rationale for the proposed system.

4. Proposed Approach section
a. Line No. 204 states that 10 best performing features from each method are carried forward to the next module. Reason for this threshold is overlooked.
b. Similarly, the feature embedding function for DNDF is missed out.
c. The statement “house probability distribution for each class” in Line No. 236 is confusing.
d. The authors are advised to provide implementation details about information gain for decision tree as well are advised to provide pseudocode or algorithm to show the integration of convolutional networks and decision tree
e. Advised to avoid too long sentences. For ex. Line No. 195 to Line No. 205 is too long to understand

Experimental design

1. Implementation details about the input and output dimension of the proposed DNDF architecture as well the training process are not discussed. This confuses the reader about the feature embedding process and the integration between feature selection and CNN module
2. The authors are advised to conduct Ablation experiments to validate the integration of feature selection method, CNN and decision tree

Validity of the findings

1. Lack of discussion on how the experimental results support to achieve the research objectives / key contribution. Especially, how the reduction in detection time is achieved

---

## Round 0.2 · Major Revisions

Reviewer 3 has again raised major issues in the paper. I suggest authors to address them before paper can be accepted.

Reviewer 2 ·

Basic reporting

Authors have improved proposed manuscript significantly, therefore I vote for acceptance.

Experimental design

Experimental design is improved.

Validity of the findings

no comments

Additional comments

no comments

Reviewer 3 ·

Basic reporting

The authors failed to improve the readability of the manuscript as pointed below,

a) From line No. 73 -75, points out that the proposed new IDS framework by handling unusual data and feature selection will address data quality, complexity, efficient use of resources. Here, what do the authors mean by unusual data ???

b) Similarly, in introduction Line No. 78 is too abstract stating that the proposed method will provide solution to many challenges. Here, what do the authors mean by many challenges ???

c) The inconsistent usage of terminology For Ex. in line no. 231, the features are referred by the word "parameters", which will confuse the reader

d) Importantly, the authors failed to gather the appropriate scholarly sources and articulate the gap in existing literature in a clear and logical manner to justify the proposed research.

Experimental design

The methodology section has not been revised adequately in accordance to my comments. Two main unaddressed comments are pointed out here

comment 1: How is the proposed model is designed to address the gap.
Decision tree inherently prioritizes features. Similarly, CNN models are effective in feature engineering process. Despite this, the proposed framework integrates feature selection method within these models. The revised manuscript failed to address my concern in this regard with valid justification.
Further, Line No. 258 -260 states that the component DNDF will be trained with 4 different feature sets and decision module will use the results of the best performing model. Hence, it is clear that the results of 4 models are ensembled. Ensemble methods, in general, are computationally expensive.


Comment 2: Implementation details about the input and output dimensions of the proposed DNDF architecture as well as the training process are not discussed. This confuses the reader about the feature embedding process and the integration between feature selection and CNN module

Updating the proposed framework diagram, will not provide details about DNDF model architecture

Validity of the findings

comment 2: how the proposed system is validated with respect to the ablation experiments and datasets
Authors failed to validate the performance of the proposed model with ablations results rather comparison with the existing model is given in Table 8.

In addition, Result discussion section lacks comprehensive explanation of the experimental results to justify the novelty of the proposed research

Additional comments

The authors have not taken efforts to address most of the raised comments

---

## Round 0.3 · accepted · Accept

The paper is accepted. However, there are still some minor English language and spelling errors and typos, that can be corrected during proofreading.

Reviewer 2 ·

Basic reporting

Dear Authors,
thank you for revising your manuscript.

Experimental design

According to my opinion, manuscript has been sufficiently improved.

Validity of the findings

According to my opinion, manuscript has been sufficiently improved.

Additional comments

According to my opinion, manuscript has been sufficiently improved.